# Data-driven design of electrolyte additives supporting high-performance 5 V LiNi$_{0.5}$Mn$_{1.5}$O$_4$ positive electrodes

Bingning Wang[1,5], Hieu A. Doan[2,5], Seoung-Bum Son [1], Daniel P. Abraham [1], Stephen E. Trask [1], Andrew Jansen [1], Kang Xu [3] ✉ & Chen Liao [1,4] ✉

LiNi$_{0.5}$Mn$_{1.5}$O$_4$ (LNMO) is a high-capacity spinel-structured material with an average lithiation/de-lithiation potential at ca. 4.6–4.7 V vs Li$^+$/Li, far exceeding the stability limits of electrolytes. An efficient way to enable LNMO in lithium-ion batteries is to reformulate an electrolyte composition that stabilizes both graphitic (Gr) negative electrode with solid-electrolyte-interphase and LNMO with cathode-electrolyte-interphase. In this study, we select and test a diverse collection of 28 single and dual additives for the Gr∥LNMO battery system. Subsequently, we train machine learning models on this dataset and employ the trained models to suggest 6 binary compositions out of 125, based on predicted final area-specific-impedance, impedance rise, and final specific-capacity. Such machine learning-generated new additives outperform the initial dataset. This finding not only underscores the efficacy of machine learning in identifying materials in a highly complicated application space but also showcases an accelerated material discovery workflow that directly integrates data-driven methods with battery testing experiments.

The spinel-structured LiNi$_{0.5}$Mn$_{1.5}$O$_4$ (LNMO) exhibits an average lithiation/de-lithiation voltage at ~4.7 V with high specific capacity and rate capability, making it a promising candidate as positive electrode material for high-energy lithium-ion batteries (LIBs), while the absence of cobalt (Co) in it brings additional advantage considering the geopolitical as well as ethical risks associated with mining of Co. However, serious challenges also arise from the high operating voltage of LNMO, which far exceeds the stability limit of any known electrolyte. One typical example of the reaction between electrolyte and LNMO is the oxidative decomposition of ethylene carbonate (EC), a prevalent electrolyte solvent in mainstream LIB industry, which forms glycolic acid and difluorophosphoric acid (HPO$_2$F$_2$) accompanied with the reduction of transition metal cores and their concomitant dissolution. The dissolved species such as Mn(II) further engage in the cross-talk between positive electrode and negative electrode, where it deposits on negative electrode surface in either metallic or ionic form, resulting in additional capacity loss as well as cell impedance rise[1,2]. To make LNMO chemistry reversible, utilization of the electrolyte additives is the most efficient and economical approach, which, without significantly changing in the mainstream electrolyte formulation and supply chain, offers several advantages. Among them are low cost, direct interphasial engineering, and minimized side effects on other important properties of electrolytes such as ion transport, chemical compatibility with other cell parts, as well as the viscosity and rheology that are already integrated as part of the mature LIB manufacturing protocol. However, the massive chemical space of electrolyte additives together with long cycling experiments often render any large-scale screening effort practically impossible.

Machine learning (ML) has rapidly become a paradigm in the field of materials science, offering acceleration in materials discovery and

[1]Chemical Sciences and Engineering Division, Argonne National Laboratory, 9700 South Cass Avenue, Lemont, IL 60439, USA. [2]Materials Science Division, Argonne National Laboratory, 9700 South Cass Avenue, Lemont, IL 60439, USA. [3]SES AI Corps, 35 Cabot Road, Woburn, MA 01801, USA. [4]Energy Storage Research Alliance, Argonne National Laboratory, 9700 South Cass Avenue, Lemont, IL 60439, USA. [5]These authors contributed equally: Bingning Wang, Hieu A. Doan. ✉e-mail: kang.xu@ses.ai; liaoc@anl.gov

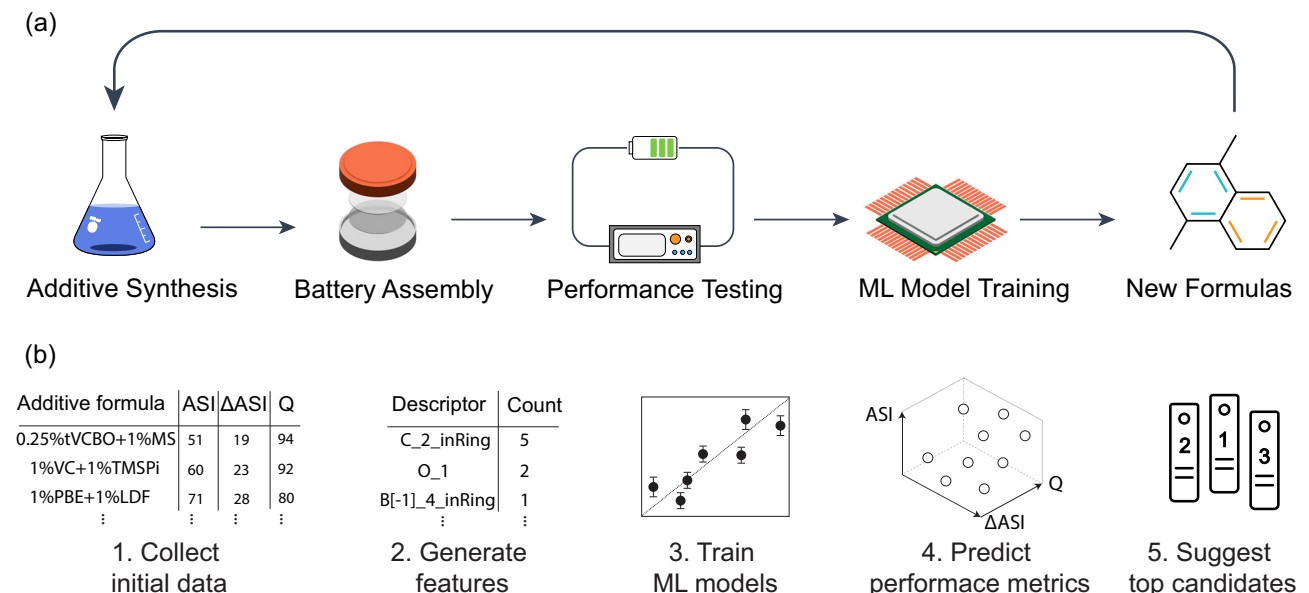

**Fig. 1 | A data-driven experimental workflow for additive discovery and optimization for lithium-ion batteries. a** Schematic representation of a machine learning (ML)-guided design of experiment workflow for electrolyte additive discovery, consisting of additive synthesis, battery assembly, performance testing, ML model training, new formula selection, and repeat. **b** Sequential method for developing ML models to predict electrolyte additives starts with initial data collection followed by feature generation and selection for ML models. Then, trained ML models are used to predict performance metrics of unknown candidates, of which the predicted top candidates are suggested for experimental validation. ASI, ΔASI, and Q denote final area-specific impedance, impedance rise, and final specific capacity, respectively.

optimization[3]. ML techniques enable the prediction of material properties, design of material structures with desired functionality, and identification of material candidates through the analysis of extensive and multifaceted datasets[4]. This approach has significantly reduced the time and cost associated with traditional experimental approaches, particularly in domains critical to technological progress such as energy storage[5] and catalysis[6]. In the realm of battery technology, the impact of ML has been profound for accelerated screening of liquid[7–9] and solid electrolytes[10–12]. Specifically, ML algorithms have been used to predict the redox potentials of electrolyte additives[13,14], as well as Coulombic efficiency[15] and cycle life[16] of LIBs as a function of additive formulas, facilitating the identification of compounds that could lead to better battery performance.

In this study, we proposed employing machine learning predictions of key battery's performance metrics to accelerate the discovery of optimal electrolyte additives. The overall ML-guided experimental workflow is illustrated in Fig. 1a, and stepwise ML tasks are shown in Fig. 1b. We initially compiled a diverse collection of electrolyte additives and examined their impact on the performance of Gr‖LNMO cells, accompanied by extensive characterizations (Fig. 1b, Step 1). A specific protocol and materials (Supplementary Note 1 and 2) were used for the testing. Artifacts such as the delamination issue are ruled out. (Supplementary Fig. 1). The resulting dataset includes the electrochemistry of Gr‖LNMO (Supplementary Note 3 and Supplementary Fig. 2–3) and allows us to explore the structure-property relationship between additives and three key performance indicators of the battery, namely the final area-specific impedance (ASI), impedance rise (ΔASI), and final specific capacity (Q). Among these, lower ASI and ΔASI indicate higher power density, improved charging/discharging rate, and enhanced efficiency, whereas a higher Q represents higher energy. Although other parameters, such as initial capacity ($Q_{ini}$) and Coulombic efficiency (CE) during cycling, can also indicate battery performance, ASI, ΔASI, and Q are the dominant factors showing the effect of additives on high-performance lithium-ion batteries[17,18]. Furthermore, we believe that leveraging these three objectives provides an effective compromise between optimization efficiency and practical

utility, ensuring a robust approach to additive discovery. We then trained and evaluated ML models with the dataset so that they can predict ASI, ΔASI, and Q based on chemical formulas and compositions of additives (Fig. 1b, Steps 2 & 3). Finally, we applied the trained models on an unknown set of 125 dual additives, of which the predicted metrics were utilized to determine the most promising candidates for experimental validation (Fig. 1b, Steps 4 & 5).

## Results and discussion

Our approach started with collecting a diverse set of additives that have been reported in the literature. These additives have been shown to contribute to the improved performance of positive and negative electrodes by reducing impedance, preventing lithium inventory loss, and mitigating electrolyte hydrolysis. In this paper, the beneficial additives for positive electrodes are referred to as positive electrode additives, while those benefitting negative electrode are referred to as negative electrode additives. The baseline solvent is 1.0 M $LiPF_6$ in ethylene carbonate (EC)/ethylmethyl carbonate (EMC) at 1/9 volumetric ratio, whose performance will be used as a reference. In our list, there are 14 positive electrode and 10 negative electrode additives (Fig. 2). The most commonly used positive electrode additives include lithium difluorophosphate (LDF)[19], in situ generated lithium malonato tetrafluorophosphates (MS)[20], and aged trimethylsilyl phosphite (TMSPi)[21]. Similarly, negative electrode additives comprises of several typical choices including lithium difluorooxalato borate (LiDFOB)[22], vinylene carbonate (VC)[23], phenylboronic acid 1,3-propanediol ester (PBE)[24], trivinylcyclotriboroxane pyridine complex (tVCBO), etc[17]. Overall, their chemical structures consist of up to seven different elements, namely C, H, Li, P, F, O, and Si. In addition, various functional groups are present in these additives, including phenyl ($C_6H_5$), phosphite (($XO)_3P$), phosphate (($XO)_3P=O$), P-F, malonato (-O-C(=O)-$CH_2$-C(=O)-O-), trimethylsilyl (-Si($CH_3)_3$), carboxyl (-C(=O)-O-), B-F, B-O, B-C, and alkene (-C=C-).

From our collection of negative electrode and positive electrode additives, we further curated and tested 10 single and 18 dual additive systems of various weight percentages (wt%). In this work, the dual

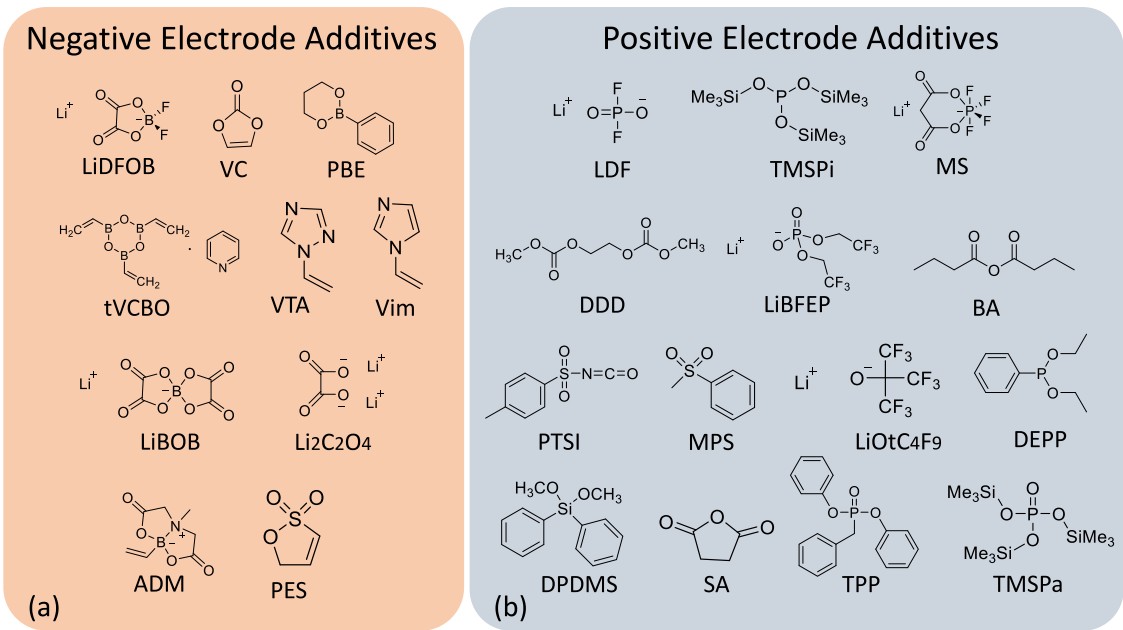

**Fig. 2 | Identity of additive candidates and their two-dimensional molecular structures. a** Additive candidates for the negative electrode. **b** Additive candidates for the positive electrode[34–42]. Their chemical names and acronyms are listed in "Methods".

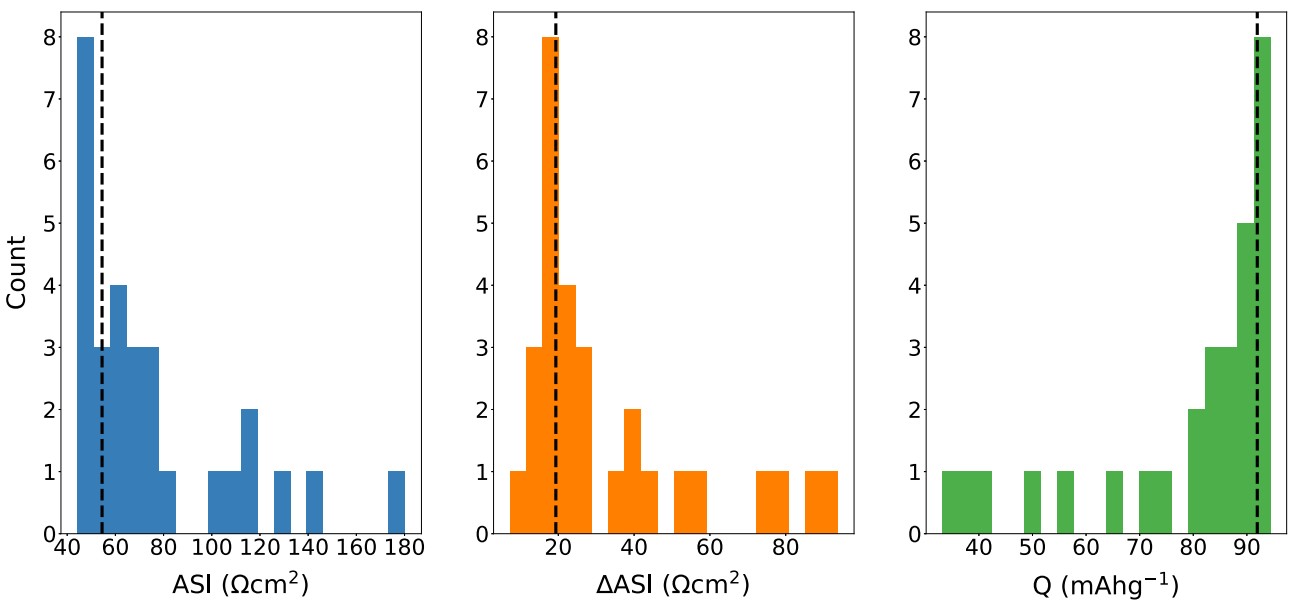

**Fig. 3 | Analysis of additive performance data.** Distributions of measured final area-specific impedance (ASI), impedance rise (ΔASI), and final specific capacity (Q) of the initial dataset of 28 additives and the baseline electrolyte. The black dashed lines indicate the measured values of the baseline solvent. Source data are provided as a Source data file.

additives always consist of a positive electrode additive and a negative electrode additive, as we hypothesized that their co-existence in the electrolyte and the synergistic effects would be critical to stabilize the two electrodes at their respective extreme potentials simultaneously. The distributions of ASI, ΔASI, and Q corresponding to 28 additives, as well as the baseline electrolyte solvent are shown in Fig. 3 (tabulated data in Supplementary Table 1). In general, these distributions were found to have non-normal trends, skewing either to the left (ASI and ΔASI) or to the right (Q) of their respective range of values. Although multiple additives contribute to improvement over the baseline in one or two performance metrics, only two dual additives, specifically tVCBO at 0.25 wt% and MS at 1.0 wt%, and LiDFOB at 1.0 wt% and TMSPi

at 1.0 wt%, surpass the baseline across all three evaluated metrics, achieving lower ASI and ΔASI, as well as a higher final specific capacity. It is also noted that many additives containing tVCBO or LiDFOB show enhanced capacity retention compared to the baseline system (Supplementary Fig. 2).

Identifying the structure-property relationships of additives is critical for understanding the impacts (whether positive or negative) of structural features/descriptors on certain targeted properties. The assignment of the descriptors/features for additives here is inspired by the previous work of Okamoto et al.[14], wherein the frequency/count of each atom and its coordination in the structure was tabulated. To further distinguish atoms beyond their coordinations, we also

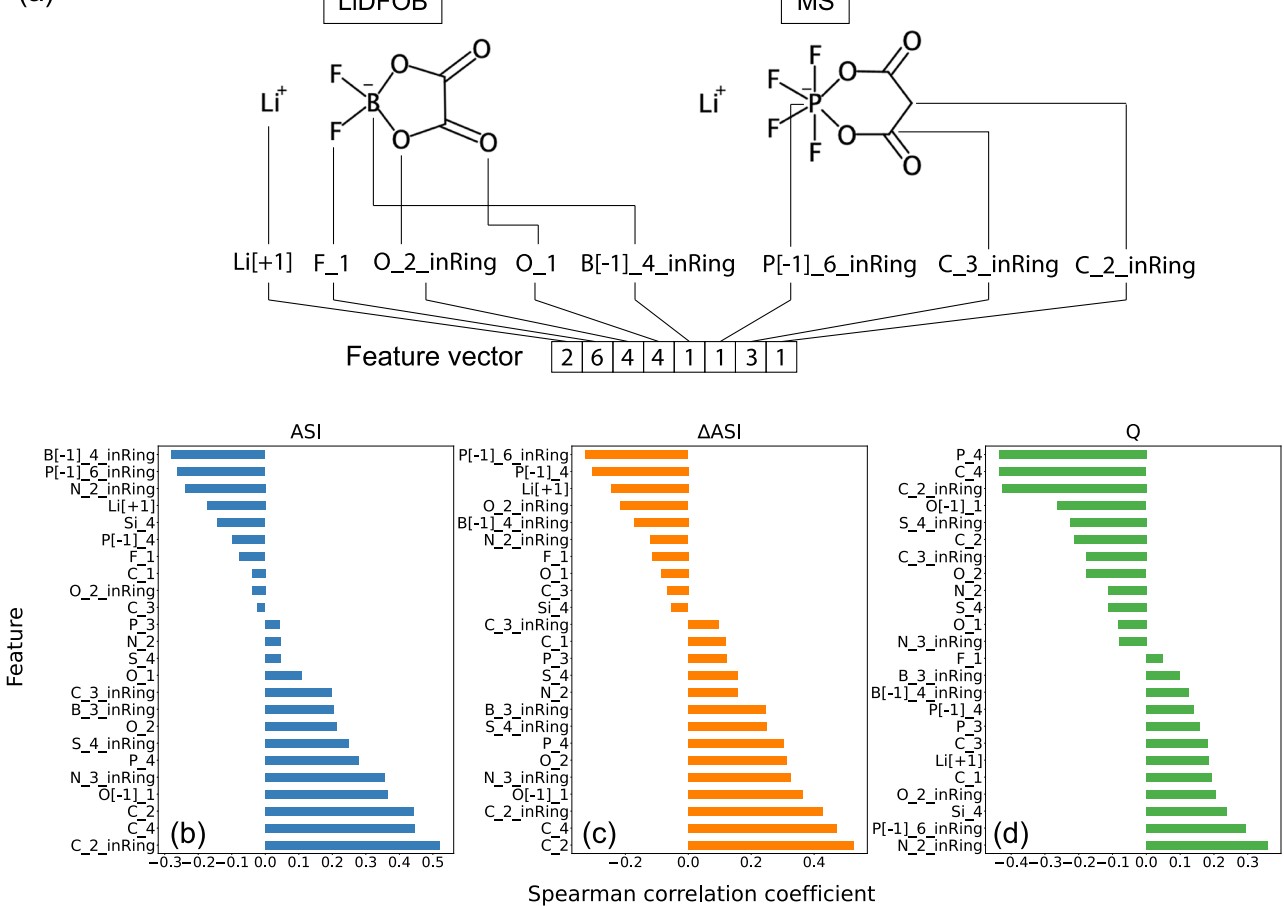

**Fig. 4 | Feature generation and analysis. a** Feature generation scheme used for molecular additives in the work. A feature vector is generated by accounting for the frequencies of unique atomic identifiers in the molecular structure of an additive. The number in the square brackets, after the underscore, and 'inRing' indicates the formal charge, the number of bonded neighbors (not including H), and whether the atom is part of a ring, respectively. Spearman correlation analysis of various features with respect to **b** final area-specific impedance (ASI), **c** impedance rise (ΔASI), and **d** final specific capacity (Q). Source data are provided as a Source data file.

incorporated additional physicochemical properties such as formal charge and whether the atom is part of a ring (Fig. 4a). For example, the feature B[-1]_4_inRing can be explained as follows: B represents the element Boron, [-1] indicates the formal charge of -1, the number 4 after the underscore is the coordination (number of neighboring atoms except for H), and "inRing" indicates that the atom B is part of a ring. This approach allows us to effectively capture the structural diversity of additives while maintaining interpretability, which is particularly important given the small size of the dataset. Additionally, as demonstrated in our benchmarking analysis (Supplementary Note 4 and Supplementary Table 2), this feature set offers a good balance between model accuracy and complexity, making it well-suited for identifying structure-property relationships in the given context. Note that the descriptor values, or the counts of distinct atomic features, were normalized to account for various concentrations of the additives. A full list of generated features and their calculated values corresponding to 28 electrolyte additives in the initial dataset is provided in the SI (Supplementary Table 3).

By analyzing the correlations between descriptors and performance metrics, we can extract the influence of each descriptor systematically. In this work, we utilized Spearman correlation analysis which describes how well the relationship between feature and performance metric can be described as a monotonic function. The most relevant features, based on Spearman correlation coefficients, with respect to ASI, ΔASI, and Q are shown in Fig. 4b–d, respectively. In these plots, positive and negative monotonic trends between each

feature and the performance metrics are indicated by positive and negative values. Notably, among the most negatively correlated features (Spearman correlation coefficient < −0.2) of additives with respect to impedance include B[-1]_4_inRing, P[-1]_6_inRing, Si_4, and N_2_inRing, respectively. Indeed, the additive combination of 1% LiDFOB and 1% MS (Fig. 4a), where both B[-1]_4_inRing and P[-1]_6_inRing features are present, has the lowest measured impedance (44.21 Ωcm²). Furthermore, these findings align remarkably well with our current knowledge of additive effects on battery performance: (1) B[-1]_4_inRing implies that the chemical structures of lithium bisoxolatoborate (LiBOB) and lithium difluorobisoxolatoborate (LiDFOB) serve as beneficial cathode electrolyte interphase (CEI) agents[25]. (2) P[-1]_6_inRing suggests that oxyfluorophosphate-based positive electrodes are favorable for low resistance and robust CEI formation[26,27]; (3) Si_4 indicates that the presence of a scavenging group, such as trimethylsilyl, effectively reduces impedance[21,26]; (4) N_2_inRing suggests that a basic group like pyrrole or morpholine behaves as an HF scavenger, reducing transition metal (TM) dissolution[26,27]. These empirical results further reinforce our design principles for positive electrode additive, demonstrating the consistency between the observed correlations in this work and the previous research findings.

As illustrated in Fig. 4c, a similar trend was observed in the Spearman correlation of the descriptors with impedance rise, with slight difference observed in some features of P[-1]_4, and O_2_inRing. The P[-1]_4 feature is associated with oxyfluorophosphate such as

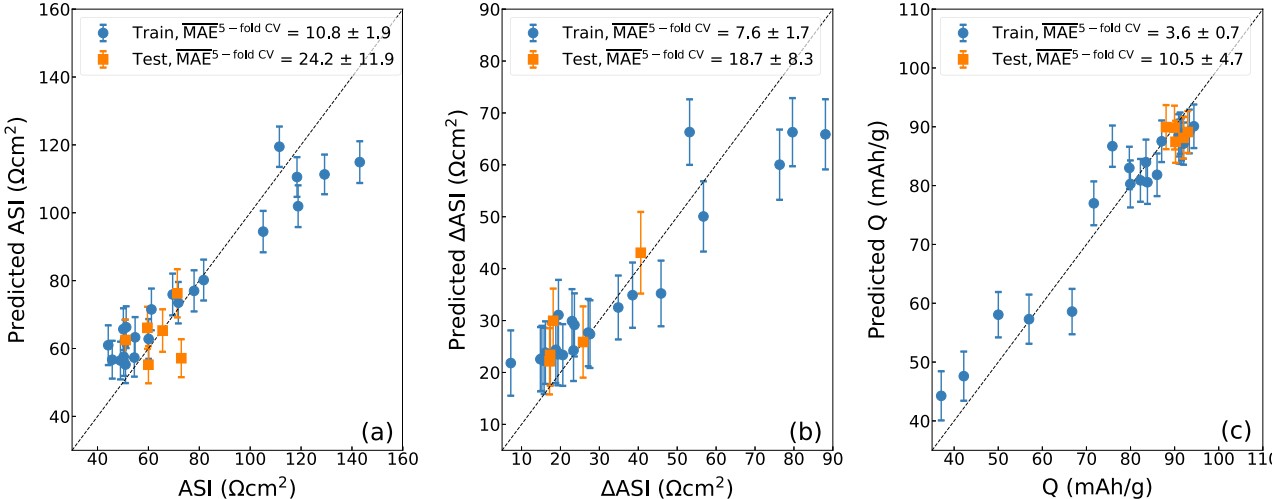

**Fig. 5 | Gaussian process regression (GPR).** Parity plot of GPR-predicted **a** final area-specific impedance (ASI), **b** impedance rise (ΔASI), and **c** final specific capacity (Q) versus measured values for the training set of 28 additive and baseline solvent. The average mean absolute values (MAEs) based on 5-fold cross validation are shown in the legends. Data are presented as mean values (solid circles and squares) +/− one standard deviation (error bars). Blue circles and orange squares indicate to train and test datapoints, respectively. Source data are provided as a Source data file.

LiPO$_2$F$_2$ and LiPO$_3$F, while O_2_inRing, in this case, is associated with boroxane structure such as in tVCBO and PBE. The correlation between the features and final specific capacity (Fig. 4d) is less insightful, as it is influenced by various interplaying factors of transition metal dissolution, lithium inventories loss, impedance, and solid-electrolyte-interphase (SEI) robustness. Nevertheless, we still can obtain some general features that carry certain chemistry significance, for example, the features that are most positively related to final specific capacity coincide with those that are inversely related to final impedance, such as P[-1]_6_inRing, B[-1]_4_inRing, and N_2_inRing. This suggests that these features are desired as they lead to both specific capacity improvement and reduction in impedance rise.

To accelerate the search for additives, it is essential to develop predictive capability ahead of tedious experiments, which typically require several months to complete. Hence, we utilized the above initial dataset to train ML models to predict potential chemical structures and compositions that could lead to improvements in ASI, ΔASI, and Q metrics. Specifically, Gaussian Process Regression (GPR) is the ML model of choice as it has been shown to be one of the most reliable algorithms for low-dimensional and small datasets[28], which is the case in this work. For our additive dataset, GPR demonstrated comparable accuracy to Random Forest regression while outperforming gradient boosting and linear regression (Supplementary Note 5). Additionally, it provides uncertainty quantification for each prediction, enabling quality evaluation of the predictions (Supplementary Table 4). We first conducted a convergence analysis to determine the optimal train/test split ratio. The resulting learning curves (Supplementary Fig. 4) indicate that an 80:20 split is optimal. To further enhance the assessment of our models' reliability, we implemented 5-fold cross-validation (CV), in which the dataset is partitioned into 5 equal segments. During each iteration, one segment is reserved for testing while the remaining nine are utilized for training. This procedure is conducted five times, with each iteration featuring a distinct test set. In addition, we repeated 5-fold CV for a total of 10 times, each with a different random state. The overall error is determined by averaging the errors across all ten models. For all models, mean absolute error (MAE) is employed as the evaluation metric. The parity plots comparing GPR predictions with experimental measurements of 28 additive and the baseline systems are shown in Fig. 5. Based on the results, the highest prediction accuracy is observed for final specific capacity model (average test MAE$^{5\text{-fold}}$

CV = 10.5 ± 4.7 mAhg$^{-1}$), followed by impedance rise model (average test MAE$^{5\text{-fold}}$ CV = 18.7 ± 8.3 Ωcm$^2$) and final area-specific impedance model (average test MAE$^{5\text{-fold}}$ CV = 24.2 ± 11.9 Ωcm$^2$)." Overall, we believe that our ML models are reasonably accurate given the size of the current training dataset. To better understand the impact of each feature on model predictions, we performed feature importance analysis using Shapley Additive Explanation (SHAP)[29] The SHAP analysis revealed the contribution of the most important features to the predictions of ASI, ΔASI, and Q, providing insights that aligned well with the trends observed in the Spearman correlation analysis (Supplementary Fig. 5). For instance, in predicting ASI, the C_2_inRing feature is positively correlated with the model output, while the B[-1]_4_inRing and P[-1]_6_inRing features have a negative impact on the predictions. This further supports the interpretability and reliability of our models.

To identify additives with improved performance, we strategically examined every possible combination of dual additives, totaling 140 pairs, by mixing 14 positive electrode and 10 negative electrode additives in equal weight percentages of 1%. Among these, 15 have already been tested and included in the initial dataset, which leaves 125 additive combinations yet to be explored. We generated the unknown compositions in this way for two reasons: (1) alignment with the training dataset, in which 15 out of 28 combinations consisted of both additives at 1%, minimizing extrapolation in the composition space; and (2) empirical evidence indicating that 1% is an effective concentration for various negative electrode and positive electrode additives[17]. We noted that brute-force experimental screening of all 125 additive combinations would be highly inefficient in terms of both time and cost (Supplementary Note 6). Using our trained GPR models, we performed prediction of ASI, ΔASI, and Q for 125 unknown additive candidates (see Supplementary Table 5 for tabulated results). Principal component analysis suggests that the model trained on the initial 28 additives adequately represents the feature space, allowing for reliable predictions for most of the 125 unknown additive systems (Supplementary Fig. 6). Furthermore, as the accuracy of GPR models has been shown to be more reliable for the prediction of ΔASI and Q with lower MAEs and uncertainty (Fig. 5), we employed those as the ranking criteria for selecting additive combinations for experimental validation. Although our approach focused on exploitation, the top candidates identified were largely consistent with those obtained through

**Table 1 | List of machine learning-suggested additives and their measured performance metrics**

| No. | Additive | ASI ($\Omega cm^2$)[a] | $\Delta$ASI ($\Omega cm^2$)[b] | Q (mAhg$^{-1}$)[c] |
|---|---|---|---|---|
| 29 | LiDFOB@1.0 wt%+SA@1.0 wt% | 41.19 | 16.26 | 91.79 |
| 30 | LiBOB@1.0 wt%+TMSPi@1.0 wt% | 104.76 | 65.57 | 69.80 |
| 31 | LiBOB@1.0 wt%+MS@1.0 wt% | 50.13 | 16.77 | 92.17 |
| 32 | LiBOB@1.0 wt%+SA@1.0 wt% | 46.79 | 10.19 | 95.49 |
| 32 | VC@1.0 wt%+SA@1.0 wt% | 123.53 | 85.94 | 47.07 |
| 33 | LiBOB@1.0 wt%+LDF@1.0 wt% | 203.43 | 110.36 | 80.06 |
| | Baseline | 54.47 | 19.35 | 91.94 |

The baseline solvent (1.0 M LiPF$_6$ in EC/EMC at 1/9 volumetric ratio) is included for reference.
[a]Area-specific impedance.
[b]Impedance rise.
[c]Final specific capacity.

Bayesian optimization methods (Supplementary Note 7 and Supplementary Table 6). The experimental measurements for the top 6 dual additives candidates are reported in Table 1, Supplementary Figs. 7 and 8, where we identified three out of six dual additives with desirable measured performance metrics (No. 29, 31, and 32). Among these, the dual system comprising of LiDFOB at 1.0 wt% and SA at 1.0 wt% shows similar Q but improved (lower) ASI and $\Delta$ASI compared to the baseline solvent. More importantly, the addition of either MS or SA to LiBOB show notable enhancement in all three considered metrics, with the combination of LiBOB at 1.0 wt% and SA at 1.0 wt% achieving the highest final specific capacity (95.49 mAhg$^{-1}$) among all additives in this work.

To gain further insights into additive performance, we carried out an array of experimental and post-test analysis of the cycled cells of the top four additive compositions and the baseline in this study, particularly focusing on the degradation mechanisms, including the regular checkup on the cycled negative electrodes for the TM cross walked from the positive electrodes (Supplementary Note 8). A specific protocol was implemented to investigate the amount of lithium inventory trapped in the negative electrode (Supplementary Fig. 9). The initial results of the kinetically accessible Li inventory obtained by electrochemical cycling are shown in Supplementary Table 7, which confirms that the difference observed in the performance is mainly due to the loss of inventory in the positive electrode that is exacerbated by transition metal (TM) dissolution. We also conducted experiments to gain insights into lithium inventory loss by performing slow cycling tests (C/100 or 17.4 $\mu$A) on reassembled cells containing aged positive electrodes and a fresh lithium chip. As depicted in Supplementary Fig. 10, the results demonstrate that the aged positive electrode that is coupled with fresh Li (blue) exhibited high specific capacities, surpassing the specific capacities of the last slow C/25 (69 $\mu$A) reference performance test cycles. This observation indicates that the degradation mechanism is primarily attributed to lithium inventory loss rather than the loss of positive electrode active materials. The performance of other cells utilizing additives, such as 1 wt% LiDFOB + 1 wt% TMSPi and 1 wt% LiBOB + 1 wt% SA, is shown in Supplementary Fig. 10.

$^1$H nuclear magnetic resonance spectroscopy clearly shows the inhibition of transesterification in presence of the designed additive combinations (Supplementary Fig. 11). X-ray photoelectron spectroscopy (XPS) confirms the formation of oxyfluorophosphates in some additives that improves CEI (Supplementary Fig. 12). Inductive coupled plasma-mass spectrometry (ICP-MS, Supplementary Fig. 13) confirms the beneficial effects of additives in reducing transition metal dissolution/deposition on the negative electrode side. SEM confirms the presence of TM aggregates in cells with additives, thereby reducing their detrimental effects on SEI rejuvenation and lithium inventory consumption (Supplementary Figs. 14–18). The online electrochemical mass spectrometry experiments have shown that these additives can also inhibit the consecutive breakdown and reformation of SEI, a process that leads to lithium inventory consumption (Supplementary Fig. 19). The experiments on harvested cell components clearly identify the lithium inventory loss as the main degradation mechanism, associated with TM dissolution and high impedance rise. All these point to the effective mitigation of degradation by these ML-predicted additive formulations.

In summary, we successfully showcased a data-driven experimental framework aimed at fast and efficient identification of electrolyte additives for LIBs based on LNMO positive electrodes. This method utilized a limited set of initial experimental data to develop reliable machine learning models that directed subsequent experimental efforts. We began by creating an initial dataset from performance metrics including final area-specific impedance, impedance rise, and final specific capacity, gathered from 28 additives and the baseline solvent. Utilizing this data, we employed ML models to evaluate these performance metrics for an expanded, untested group of 125 dual electrolyte additives. Remarkably, by experimentally validating only the top 6 candidates identified through ML predictions, we discovered a binary formulation, namely LiBOB at 1 wt% and SA at 1 wt%, that outperformed all additives in the initial dataset. Future research will explore a broader array of additives, including ternary compositions, via closed-loop experiments guided by Bayesian optimization. The methodology described herein has the potential to be applied universally to other areas of materials discovery, particularly where navigating vast design spaces and conducting time-intensive experiments are major hurdles.

## Methods
### Selection of additive candidates
A series of additives, commonly employed to enhance the performance of either positive electrodes or anodes, have been utilized in the study. These additives are known to have beneficial effects such as reducing impedance, preventing lithium inventory loss, and mitigating electrolyte hydrolysis. They have been traditionally reported to contribute to the improvement of positive and negative electrode performance by addressing these specific issues. As depicted in Fig. 2, both the single and dual additives (combination of two additives), are employed in the positive and negative electrode systems. For the positive electrode additives, we have included the following: lithium difluorophosphate (LDF)[19], in situ generated lithium malonato tetrafluorophosphates (MS)[20], and aged trimethylsilyl phosphite (TMSPi)[21]. The aged TMSPi refers to a static aging process of approximately 1 week to allow the spontaneous reaction between the LiPF$_6$ and TMSPi to reach optimal concentration of (OTMS)$_y$PF$_x$ to serve as a positive electrode additive[21,30]. In terms of the negative electrode additives, our selection comprises lithium difluorooxalato borate (LiDFOB)[22], vinylene carbonate (VC)[23], and phenylboronic acid 1,3-propanediol ester (PBE)[24], and trivinylcyclotriboroxane pyridine

**Table 2 | Composition of the laminates used in this work**

| Positive electrode: LN210O035-179-1 | Negative electrode: A-A015A |
|---|---|
| (single-sided) | (single-sided) |
| 90 wt% Targray LNMO "5V spinel" | 91.83 wt% Superior Graphite SLC1506T |
| 5.69 wt% Timcal C-45 | 2 wt% Timcal C-45 Carbon |
| 0.05 wt% Tuball SWCNT | 6 wt% TKureha 9300 PVDF Binder |
| 4.26 wt% solvay 5130 PVDF Binder | 0.17 wt% Oxalic Acid |
| SS- single sided → calendered | SS- single sided → calendered |
| Total Electrode Thickness 83 µm | Total Electrode Thickness 53 µm (ss) SS coating thickness: 47 µm |
| Porosity: 40.9% | Porosity: 37.5% |
| Total SS Coating Loading 15.21 mg/cm$^2$ | Total SS Coating Loading 6.37 mg/cm$^2$ |
| Total SS Coating Density, 2.31 g/cm$^3$ | Total SS Coating Density, 1.36 g/cm$^3$ |
| Estimated C/10 areal capacity 1.85 mAh/cm$^2$ | Estimated C/10 areal capacity 1.93 mAh/cm$^2$ |

complex (tVCBO)[17]. The selection represents different elements consisting of C, H, Li, P, F, O, Si. Several functional groups are also represented in these additives, including aromatic of $C_6H_5$-, and $C_5H_5$ groups, P=O, P-F, -O-C(=O)-$CH_2$-C(=O)-O-, -$SiMe_3$, -C(=O)-O-, B-F, B-O, B-$CHCH_2$, -C=C-, etc.

The structures of the additives are shown in Fig. 2, and their acronyms and full names are listed below:

LiDFOB, lithium difluorobisoxalatoborate; VC, vinylene carbonate; PBE, Phenylboronic acid 1,3-propanediol ester; tVCBO, trivinylcyclotriboroxane; VTA, 1-vinyl-1,2,4-triazole; Vim, 1-Vinylimidazole; LiBOB, Lithium bis(oxalato)borate; $Li_2C_2O_4$, Lithium oxalate; ADM, 6-Methyl-2-vinyl-1,3,6,2-dioxazaborocane-4,8-dione; LDF, lithium difluorooxyphosphate; TMSPi, tris(trimethylsilyl) phosphite; MS, in situ lithium malonatotetrafluorophosphate; DDD, dimethyl-2,5-dioxahexanedioate; LiBFEP, lithium bis(2,2,2-trifluoroethyl) phosphate; BA, Butyric anhydride; PTSI, p-toluenesulfonyl; MPS, Methyl phenyl sulfone; $LiOtC_4F_9$, lithium nonafluoro-tert-butoxide; DEPP, diethyl phenylphosphonite; DPDMS, dimethoxydiphenylsilane; SA, succinic anhydride; TPP, triphenyl phosphate; TMSPa, tris(trimethylsilyl) phosphate; PES, prop-1-ene-1,3-sultone.

## Positive electrodes, negative electrodes, separators, and coin cells

The Ni-rich positive electrode laminates and Gr negative electrode laminates used in this study were supplied by Argonne's Cell Analysis, Modeling, and Prototyping facility. The laminates were coated by an automatic slot die coater in a dry room and dried under vacuum at 80 °C. They were also dried at 100 °C under vacuum in Ar-filled glovebox right before use. Detailed information about the composition of laminates is provided in Table 2. The coin cell assembly utilized the microporous separator Celgard 2325. Prior to use, all electrodes were dried at 110 °C under vacuum in an argon-filled glovebox, while the coin cell components (excluding the separators) were dried in an oven at 100 °C. The separators were dried at 50 °C overnight. The 2032-type coin cells were prepared within an Ar-filled glovebox. The diameters of the positive electrode, graphite electrode, and separator were chosen to be 14, 15, and 16 mm, respectively. The discs were cut using Nogami Scissor Punches. The total amount of added electrolyte was 25 µl per cell. For each electrolyte formulation, three individual cells were tested. A CR2032 coin cell (www.predmaterials.com) and a VWR® Signature™ Ergonomic High Performance Single Channel Variable Volume Pipettors were used together with VWR tips. Extra wetting is not required for the assembled cells. We made all cells in triplicate and kept them in an environmental chamber (convection heating) at a constant temperature of 30 °C.

## $^1$H Nuclear magnetic resonance (NMR) spectroscopy

The aged cells were opened in an Ar-filled glove box using insulated pliers, and the electrolyte was gathered by immersing the electrodes and the separators in 1.0 ml anhydrous $CDCl_3$ for one minute. The resulting solutions were analyzed using $^1$H NMR spectroscopy. NMR spectra were obtained using a Bruker Avance III HD 300 MHz spectrometer, and the chemical shift of $CDCl_3$ at 7.26 ppm was used as a reference. Prior to measurement, all solutions were handled in an Ar-filled GB at room temperature (20 ± 3 °C).

## X-ray photoelectron spectra (XPS) characterization

The spectra were acquired with a PHI 5000 VersaProbe II System (Physical Electronics) using a base pressure of $2 \times 10^{-9}$ Torr. Prior to measurements, the aged electrodes were cleaned with dimethyl carbonate (DMC) and left inside an Ar-filled glovebox for drying. The photoelectron spectra were collected in the fixed analyzer transmission mode using an Al Kα radiation (1486.6 eV, 100 µm beam, 25 W) with Ar$^+$ ion and electron beam sample neutralization. The spectra were calibrated against the graphitic carbon at 284.5 eV. Samples were moved between gloveboxes using a Mason jar filled with argon, and the XPS system is linked to the argon-filled glovebox.

## Inductively coupled plasma-mass spectra (ICP-MS)

To quantitatively assess the transition metal dissolution in the aged cells, the cycled negative electrodes were washed, placed in a quartz beaker, and incinerated in a furnace at 700 °C for 12 h. All organic constituents and carbon were removed in this process. The resulting ash were treated with a refluxing mixture of nitric and hydrochloric acids at 220 °C for one hour, and the solutions were then treated with water. Inductive coupled plasma-mass spectrometry (ICP-MS) were used to determine the transition metal concentrations, and the weight of the negative electrode was used as a reference. Measurements were conducted with a PerkinElmer NexION 2000 ICP Mass Spectrometer calibrated using the NIST traceable standards. All samples were handled in ambient condition (air, room temperature of 20 ± 3 °C).

## Scanning electron microscopy (SEM)

The positive electrode discs were carefully washed with 1 ml of dimethyl carbonate (DMC) for 1 min then left to dry inside a glovebox overnight prior to microscopic examination. Once the cells were opened by insulated pliers in an Ar-atmosphere glovebox, aged electrodes were moved to the SEM chamber without air exposure by using portable air-lock chamber. Images and EDS spectra were obtained using a JEOL JSM-6610LV SEM coupled with an EDS detector operating at 20 kV.

## Online electrochemical mass spectrometry (OEMS)

Online electrochemical mass spectrometry (OEMS) was employed for real-time and quantitative analysis of the gaseous species produced during electrochemical experiments. The diagram of our OEMS experimental setup is shown in Fig. 6. We used an FMA-2600/FVL-2600 SERIES Mass and Volumetric instrument from OMEGA to regulate the flow rate of a He tank. The Hiden HPR-40 DEMS system featured a quadrupole mass spectrometer and a QIC UF microflow capillary inlet (type 303452) with a flow rate of 12 µL/min. A PX409-015GUSBH (Pressure Sensor, 15 psi, Digital, Gauge, 1/16 in.) transducer was utilized to monitor the real-time pressure to quantify gaseous species. The system includes five manual Swagelok ball valves (SS-41GS1) were incorporated into the system to enable gas line evacuation and manage the flow rate/testing. An ECC-DEMS cell from El-cell was used with a two-electrode configuration: LNMO served as the positive electrode and graphite as the negative electrode. Each cell contained 120 µL of electrolyte. A VSP-300 Potentiostat from Biologic was used to cycle the cells between 2.5 and 5 V for two cycles, with a 3 h-constant voltage hold at 5 V after each charge. The El-cell

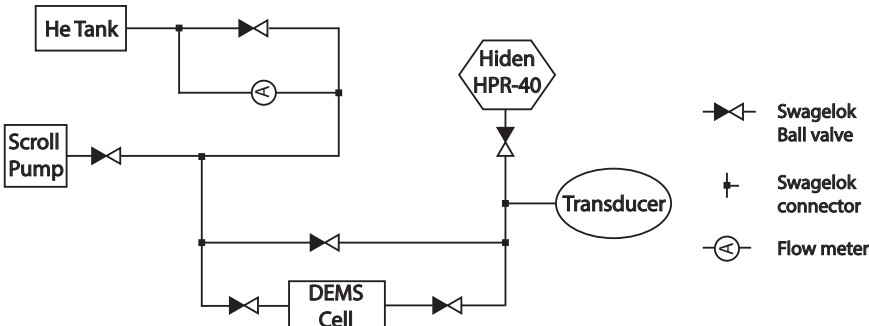

**Fig. 6 | Schematic drawing of the online electrochemical mass spectroscopy (OEMS) system used in this study.** The scheme shows that a flow meter regulated He was used as the carrier gas for the differential electrochemical mass spectroscopy (DEMS) cell, while 3 Swagelok cells as well as a bypass line is used to ensure the vacuuming and refilling of He through the whole system.

was prepared in an Ar-filled glovebox and the OEMS's unique design allows vacuuming and refilling of Ar before the cell is reconnected with the OEMS system. All measurements were conducted at a temperature of $20 \pm 3$ °C.

## Gaussian process regression

A Gaussian process (GP) is a collection of random variables, any finite number of which have a joint Gaussian distribution[31]. A GP is completely specified by its mean function m($x$) and covariance function (or kernel) k($x$,$x'$), and can be written as:

$$f(x) \sim GP(m(x), k(x, x')) \tag{1}$$

For a new input $x^*$, the predicted mean, $\mu_*$, and variance, $\Sigma_*$, of f($x^*$) are given as:

$$\mu_* = \mathbf{K}_*^T \mathbf{K}_y^{-1} y \tag{2}$$

$$\Sigma_* = \mathbf{K}_{**} - \mathbf{K}_*^T \mathbf{K}_*^{-1} \mathbf{K}_* \tag{3}$$

where y is the labeled property of the training set and $\mathbf{K} = k(\mathbf{X}, \mathbf{X})$, $\mathbf{K}_* = k(\mathbf{X}, \mathbf{X}_*)$, and $\mathbf{K}_{**} = k(\mathbf{X}_*, \mathbf{X}_*)$. $\mathbf{X}$ and $\mathbf{X}_*$ are the feature matrices of the train and test sets, respectively. If $x$ and $x'$ represent the feature vectors, then their covariance based on the Matérn kernel ($v = 1.5$) is expressed as follows:

$$k(x, x') = \left(1 + \frac{\sqrt{3}|x - x'|}{\sigma_l}\right) * \exp\left(-\frac{\sqrt{3}|x - x'|}{\sigma_l}\right) + \sigma_n^2 \tag{4}$$

Here, $\sigma_l$ and $\sigma_n$ are the length scale and the expected noise level in the data set, respectively. Each parameter was determined using the maximum likelihood estimate during model training, using Adam optimizer with a learning rate of 0.1 as implemented in the GPyTorch library[32].

## Data availability

All data generated in this study are provided in the Supplementary Information Source data file. Source data are provided with this paper.

## Code availability

Code is available on Zenodo[33] and Github repository: https://github.com/hieuadoan/ML-LNMO-additives.

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

## Acknowledgements

Support from the Vehicle Technologies Office of the U.S. Department of Energy, particularly from the Earth-abundant Cathode Active Materials (EaCAM) consortium managed by Tien Duong and Brian Cunningham, is gratefully acknowledged. C.L., B.W., S.S., D.A., S.T., and A.J. received funding from EaCAM. The electrodes and electrolytes used in this article are from Argonne's Cell Analysis, Modeling and Prototyping Facility (CAMP) and Materials Engineering Research Facility (MERF). The submitted manuscript has been created by UChicago Argonne, LLC, Operator of Argonne National Laboratory ("Argonne"). Argonne, a U.S. Department of Energy Office of Science laboratory, is operated under Contract No. DE-AC02–06CH11357.

## Author contributions

C.L. and H.A.D. conceived the research. B.W., C.L., and S.E.T. designed the experiments. B.W. performed the experiments. B.W. and S.S. carried out an experimental data analysis. H.A.D. conducted feature generation and selection, and developed code for implementing machine learning algorithms. H.A.D. and C.L. wrote the manuscript. S.S., S.E.T., D.P.A., A.J., and K.X. contributed to the revision of the manuscript. C.L. provided supervision, project administration, and funding acquisition. All authors have given approval to the final version of the manuscript.

## Competing interests

The authors declare no competing interests.
