## [Peer Review File · Nature Communications]

Data-driven Design of Electrolyte Additives Supporting High-Performance 5 V LiNi_{0.5}Mn_{1.5}O₄ Positive Electrodes

Corresponding Author: Dr Chen Liao

Version 0:

Reviewer comments:

Reviewer #1

(Remarks to the Author)

The authors trained a machine learning model using 28 different single- and dual-additive experimental datasets and used the model to identify six optimal binary combinations from 125 additives. The manuscript presents a novel work to investigate electrolytes for high performance LIBs. They proposed an interesting approach of directly combining data-driven and battery-test experiments to select additives for optimized electrolytes, which showcases a significant reference to the readers in the relevant community. It may have the potential for further research and application. The code is well run reproducibly. It is recommendable that the manuscript could be published after addressing the following issues.

1. The authors used the parameters of area specific impedance (ASI), area impedance rise (Δ ASI), and final specific capacity (Q) to evaluate the additives for electrolytes design. If other parameters could be employed or the parameters affect the machine learning model? There is an inter-correlation between the three indicators (specific impedance (ASI), impedance rise (Δ ASI), and final specific capacity (Q)) chosen by the authors. Does this correlation have an impact on the model?
2. The well-formed descriptors are the basis for implementing machine learning models, and the authors specifically introduced "rings" as part of the descriptors. What are the benefits of this improved approach?
3. The authors experimentally tested the effects of 28 additives in relatively small amount. Does the machine learning model fall into local optimal solutions? How well does the model scale up to other electrolyte additive systems?
4. The authors used Gaussian Process Regression (GPR) regression method. Can other regression methods be used for training and is GPR optimal among the various methods?
5. The fitted convergence curves for machine learning need also to be provided to analyze the convergence effect.

(Remarks on code availability)

The code provided by the authors is a usable resource and includes a README file with descriptions of the various files. The code can be successfully installed and run, and the content of the manuscript on machine learning is reproducible.

Reviewer #2

(Remarks to the Author)

This study employed machine learning to identify and select optimal electrolyte additives, demonstrating superior performance and showcasing the efficacy of ML in accelerating material discovery and improving battery technology. The manuscript is in general well-written and provides valuable technological insights, but several concerns should be addressed before publication:

- The initial dataset of 28 samples is too small for training the machine learning model due to potential overfitting issues. Although 10-fold cross-validation was performed, the fixed train-test set limits the robustness of the model. It is recommended to use 5-fold cross-validation and also vary the train-test ratio from 2:8 to 8:2 with at least 10 different random states to better validate the model's performance. In addition, the performance of various types of ML algorithms should be provided.
- The feature engineering process lacks clarity. The authors should consider all possible physical, chemical, and structural features and conduct feature importance analysis using SHAP or LIME to ensure the most effective and interpretable feature set.
- The study appears to use exploitation to validate the ML model's predictive capability with 128 sets. However, Bayesian

based active learning, which is more effective and has been confirmed its validity from various researches, should be considered for the optimization process.

- The rationale for using ML in this study is unclear. While conducting 128 experiments is time-consuming and costly, the search space is relatively small from an ML perspective. The authors should justify that 128 sets are sufficient and that the best result is included. A more rigorous search and justification for the additive combinations are needed. It seems conducting 128 experiments would be more effective.
- The extrapolation performance of the ML model should be validated, as ML models are typically weak in this area. Dimensionality reduction techniques should be applied to both train (28) and test (128) sets to understand the ML search space and determine whether it involves extrapolation or interpolation.
- The authors should demonstrate that the optimal set of additives is also effective for different chemical compositions of LNMO (e.g., LiNi_{0.6}Mn_{1.4}O₄ or LiNi_{0.5}Mn_{1.4}Dopant_{0.1}O₄), as the tested LNMO composition may not be the best cathode material.

(Remarks on code availability)

It seems the results are reproducible based on the provided codes.

Version 1:

Reviewer comments:

Reviewer #1

(Remarks to the Author)

The resubmitted manuscript shows that the authors have made revisions and offered supplementary materials as much as possible in response to the comments. Overall, the response and revision are satisfactory. Nevertheless, I still have a short comment on the response to Reviewer1_Question1 for your reference as below, even though the question isn't decisive to the article acceptance.

There is a significant correlation between the selected performance indicators in the study with Pearson's correlation coefficients of 0.93, -0.79 and -0.87 for ASI/ Δ ASI, ASI/Q and Δ ASI/Q, respectively. The authors believe that the correlation indicator suggests that it is possible to consider the use of only one indicator. Although the absolute value of the coefficients is close to 1, a larger number of factors/indicators still need to be considered. We can assume that all performance indicators form a vector space and that the Δ ASI, ASI, and Q indicators are linear combinations of similar basis vectors in that space, and therefore have some correlation. However, these combinations also contain different components that play an important role in the discovery of new electrolytes, making it necessary to give due consideration to additional performance indicators.

(Remarks on code availability)

The code can be run well.

Reviewer #2

(Remarks to the Author)

The authors have adequately addressed the raised concerns; therefore, I recommend it for publication.

(Remarks on code availability)

Reviewer #1 (Remarks to the Author):

The authors trained a machine learning model using 28 different single- and dual-additive experimental datasets and used the model to identify six optimal binary combinations from 125 additives. The manuscript presents a novel work to investigate electrolytes for high performance LIBs. They proposed an interesting approach of directly combining data-driven and battery-test experiments to select additives for optimized electrolytes, which showcases a significant reference to the readers in the relevant community. It may have the potential for further research and application. The code is well run reproducibly. It is recommendable that the manuscript could be published after addressing the following issues.

We thank the reviewer for the constructive comments.

1. The authors used the parameters of area specific impedance (ASI), area impedance rise (Δ ASI), and final specific capacity (Q) to evaluate the additives for electrolytes design. If other parameters could be employed or the parameters affect the machine learning model? There is an inter-correlation between the three indicators (specific impedance (ASI), impedance rise (Δ ASI), and final specific capacity (Q)) chosen by the authors. Does this correlation have an impact on the model?

We thank the reviewer for the suggestion. Other parameters, such as initial capacity (Q_{ini}) and coulombic efficiency (CE) during cycling, also contribute to battery performance. However, ASI, Δ ASI, and Q are the key parameters predominantly influence the effects of additives on high-performance lithium-ion batteries. For example, in the previous publications from the Deep Dive Consortium of Cathodes and our recent book,¹ Figure of Merit were introduced as a quantitative measurement for performance evaluation, particularly for systems that shows subtle yet improved performance. The Figure of Merit Power was defined as the extrapolated number of reaching the 80 % of baseline power density, which was calculated through the formulation of the following:

$$\text{Discharge pulse power} = V_{min} \frac{OCV - V_{min}}{\text{Discharge resistance}}, \quad \text{Eq. (1)}$$

where discharge resistance is referring to ASI that can be calculated in the following equation

$$\text{ASI} = \frac{V_{t_0} - V_{t_1}}{I_{t_1} - I_{t_0}} \quad \text{Eq. (2)}$$

¹ J Power Sources 2017 (365), 201-209; Batteries, C. Liao ed., IOP Publishing 2021, DOI: 10.1088/978-0-7503-2682-7

Figure 1. a) Typical protocol (voltage vs. time) for testing a full cell consisting of a graphite anode and a transition metal oxide cathode; the cutoff voltages are 3 and 4.2 V, and the inset shows the voltage profile of the hybrid pulse power characterization (HPPC) cycle. b) Schematic representation of the cell voltage and current for the ASI measurement.

The FOM energy density is calculated as the extrapolation of the number of cycles at which the energy density decreases to the 80 % of the initial value (using baseline number in $\text{mWh}_{\text{oxide}}^{-1}$). This number is directly related to the final Q that is measured at C/25. As shown in the Supp. Info, a total of five Reference performance tests (RPT) of C/25 were introduced, with intermittent 19 aging cycles at C/3 between each one.

A more presentative view of how FOME and FOMP with NMC as cathodes are calculated is shown in Figure 2. Clearly, the initial ASI, final ASI, and final Q are the parameters that determine the FOMP and FOME, which has been demonstrated to be used dominantly to evaluate the performance of additives on high performance lithium-ion batteries.

Fig. 2. FOME (a) and FOMP (b) derivation schematic for the baseline electrolyte cell. For FOME, the top line indicates the Energy Density (ED) at cycle 5 (C/10), and the bottom line indicates 80% of that value. The FOME is the cycle number at which the extrapolated C/10 ED reaches that 80% threshold, which occurs at 170 cycles for the baseline. For FOMP, the top line indicates the Power Density (PD) calculated for the baseline cell at 4.08 V for the first HPPC test. The cell power is calculated from the ASI and fit to a logarithmic decay. The FOMP is the cycle number at which the fit of cell power crosses 80% of the initial baseline power, which occurs at 23 cycles for the baseline.

As noted by the reviewer, there is significant intercorrelation among the selected performance indicators in this study. Specifically, the Pearson correlation coefficients for ASI/ Δ ASI, ASI/Q, and Δ ASI/Q are 0.93, -0.79, and -0.87, respectively. Although these correlations suggest that it may be possible to consider only a single metric, such as Q, the limited amount of data led us to cautiously include all three. Since independent ML models are developed for ASI, Δ ASI, and Q, the accuracy of these models is not impacted by the correlations among the metrics.

We have added the following sentences to the manuscript to further clarify the use of ASI, Δ ASI, and Q:

“Although other parameters, such as initial capacity (Q_{ini}) and coulombic efficiency (CE) during cycling, can also indicate battery performance, ASI, Δ ASI, and Q are the dominant factors showing the effect of additives on high-performance lithium-ion batteries. (*Batteries*, 2021; Tornheim et al., 2017)”

2. The well-formed descriptors are the basis for implementing machine learning models, and the authors specifically introduced "rings" as part of the descriptors. What are the benefits of this improved approach?

We appreciate the reviewers for highlighting this point. We introduced the 'inRing' descriptor to further distinguish between cyclic and acyclic atoms, which may otherwise have identical descriptors (e.g., C₂ vs. C₂_inRing, O₂ vs. O₂_inRing, S₄ vs. S₄_inRing). This

differentiation allows for a more precise identification of the relationships between performance metrics and these descriptors. For instance, while O₂ shows a positive Spearman correlation with ASI and Δ ASI and a negative correlation with Q, O₂_inRing exhibits the opposite trends (Fig. 4 in the manuscript).

3. The authors experimentally tested the effects of 28 additives in relatively small amount. Does the machine learning model fall into local optimal solutions? How well does the model scale up to other electrolyte additive systems?

We agree with the reviewer that the number of experimentally tested additives is limited. With only 28 data points, our trained ML models may extrapolate when making predictions on the 125 unknown additive systems. To further investigate this, we conducted a principal component analysis (PCA) on all considered additives and projected their features onto the first two principal components, as shown in Fig. S19. The results indicate that the features of the initial 28 additives are diverse enough to overlap with approximately 85 out of the 125 unknown additives, suggesting reasonable coverage of the feature space. This overlap provides confidence that the model, trained on the initial 28 additives, has sufficient representation of the feature space to make reliable predictions for the majority of the 125 unknown additive systems.

We have modified the manuscript accordingly to reflect this observation:

“Principal component analysis suggests that the model trained on the initial 28 additives adequately represents the feature space, allowing for reliable predictions for most of the 125 unknown additive systems (Fig. S19)”

Figure S19 Graphical illustration of the feature space of all considered additives projected onto principal component 1 and 2.

4. The authors used Gaussian Process Regression (GPR) regression method. Can other regression methods be used for training and is GPR optimal among the various methods?

In addition to GPR, we evaluated the performance of several regression methods, including linear regression, gradient boosting, and random forest. To ensure robust results, we repeated the model training and testing process 100 times for each method, reporting the means and standard deviations of the test mean absolute error in Table S4. Our findings indicate that GPR and Random Forest Regression provide comparable prediction accuracy, both outperforming gradient boosting (by a small margin) and linear regression (by a significant margin).

We have added the following sentence to the manuscript to further clarify the use of GPR: “For our additive dataset, GPR demonstrated comparable accuracy to Random Forest Regression while outperforming gradient boosting and linear regression (Table S4). Additionally, it provides uncertainty quantification for each prediction, enabling quality evaluation of the predictions (see Machine Learning methods in the SI)”

Table S4 Prediction accuracy of different ML algorithms for ASI, Δ ASI and Q. Training and testing were repeated 100 times using different random states.

Test MAE*	Linear Regression	Gradient Boosting	Random Forrest Regression	Gaussian Process Regression
ASI	39.9 \pm 36.6	22.0 \pm 9.0	20.4 \pm 7.3	20.2 \pm 6.9
Δ ASI	20.7 \pm 17.4	16.3 \pm 6.3	15.6 \pm 5.5	16.7 \pm 6.1
Q	14.4 \pm 10.3	12.9 \pm 5.2	12.0 \pm 4.1	12.2 \pm 4.3

*Train:test = 80:20

5. The fitted convergence curves for machine learning need also to be provided to analyze the convergence effect.

We appreciate the reviewer’s suggestion. We have plotted the learning curves to analyze the convergence effect of different train/test split ratios. As shown in Fig. S17, an optimal train:test split of approximately 80:20 was identified, supporting our use of 5-fold cross-validation for assessing model performance.

Figure S17 Prediction accuracy of GPR models evaluated at different train/test split ratios, averaged over 100 runs.

We have included the analysis of the fitted convergence curves in the SI. The following sentence is also added to the manuscript:

“We first conducted a convergence analysis to determine the optimal train/test split ratio. The resulting learning curves (Fig. S17) indicate that an 80:20 split is optimal”

Reviewer #1 (Remarks on code availability):

The code provided by the authors is a usable resource and includes a README file with descriptions of the various files. The code can be successfully installed and run, and the content of the manuscript on machine learning is reproducible.

Reviewer #2 (Remarks to the Author):

This study employed machine learning to identify and select optimal electrolyte additives, demonstrating superior performance and showcasing the efficacy of ML in accelerating material discovery and improving battery technology. The manuscript is in general well-written and provides valuable technological insights, but several concerns should be addressed before publication:

1. The initial dataset of 28 samples is too small for training the machine learning model due to potential overfitting issues. Although 10-fold cross-validation was performed, the fixed train-test set limits the robustness of the model. It is recommended to use 5-fold cross-validation and also vary the train-test ratio from 2:8 to 8:2 with at least 10 different random states to better validate the model's performance. In addition, the performance of various types of ML algorithms should be provided.

In response to the reviewer's suggestion, we have updated the performance analysis of the ML models using 5-fold cross-validation with 10 different random states. The revised results are now presented in Fig. 5 of the manuscript.

We have also updated the texts in the manuscript accordingly:

“To further enhance the assessment of our models' reliability, we implemented 5-fold cross-validation (CV), in which the dataset is partitioned into 5 equal segments. During each iteration, one segment is reserved for testing while the remaining nine are utilized for training. This procedure is conducted five times, with each iteration featuring a distinct test set. In addition, we repeated 5-fold CV for a total of 10 times, each with a different random state. The overall error is determined by averaging the errors across all ten models. For all models, mean absolute error (MAE) is employed as the evaluation metric. The parity plots comparing GPR predictions with experimental measurements of 28 additive and the baseline systems are shown in Figure 5. Based on the results, the highest prediction accuracy is observed for final specific capacity model (average test $MAE^{5\text{-fold CV}} = 10.5 \pm 4.7 \text{ mAhg}^{-1}$), followed by impedance rise model (average test $MAE^{5\text{-fold CV}} = 18.7 \pm 8.3 \text{ } \Omega\text{cm}^2$) and final area specific impedance model (average test $MAE^{5\text{-fold CV}} = 24.2 \pm 11.9 \text{ } \Omega\text{cm}^2$).”

Figure 5 Parity plot of GPR-predicted (a) final area specific impedance (ASI), (b) impedance rise (Δ ASI), and (c) final specific capacity (Q) versus measured values for the training set of 28 additive and baseline solvent. The average mean absolute values (MAEs) based on 5-fold cross validation are shown in the legends. The error bars indicate one standard deviation.

In addition to GPR, we evaluated the performance of several regression methods, including linear regression, gradient boosting, and random forest. To ensure robust results, we repeated the model training and testing process 100 times for each method, reporting the means and standard deviations of the test mean absolute error in Table S4. Our findings indicate that GPR and Random Forest Regression provide comparable prediction accuracy, both outperforming gradient boosting (by a small margin) and linear regression (by a significant margin).

Table S4 Prediction accuracy of different ML algorithms for ASI, Δ ASI and Q. Training and testing were repeated 100 times using different random states.

Test MAE*	Linear Regression	Gradient Boosting	Random Forrest	Gaussian Process Regression
ASI	39.9 ± 36.6	22.0 ± 9.0	20.4 ± 7.3	20.2 ± 6.9
Δ ASI	20.7 ± 17.4	16.3 ± 6.3	15.6 ± 5.5	16.7 ± 6.1
Q	14.4 ± 10.3	12.9 ± 5.2	12.0 ± 4.1	12.2 ± 4.3

* Train:test = 80:20

We have added the following sentence to the manuscript to further clarify the use of GPR: “For our additive dataset, GPR demonstrated comparable accuracy to Random Forest Regression while outperforming gradient boosting and linear regression (Table S4). Additionally, it provides uncertainty quantification for each prediction, enabling quality evaluation of the predictions (see Machine Learning methods in the SI)”

2. The feature engineering process lacks clarity. The authors should consider all possible physical, chemical, and structural features and conduct feature importance analysis using SHAP or LIME to ensure the most effective and interpretable feature set.

We thank the reviewer for the suggestion. To improve the clarity of our feature engineering process, we have benchmarked the accuracy of GPR models using additional feature sets, with the results presented in Table S3. The physicochemical feature set comprises 66 unique physical and chemical descriptors, including molecular weight, topological polar surface area, number of valence electrons, and number of aromatic rings. The extended connectivity fingerprint (ECFP) is a representation based on atom-centered circular neighborhoods, where atom environments are iteratively expanded and hashed into fixed-length binary vectors. Both the physicochemical features and ECFP can be automatically generated for each additive system using the RDKit package.

As shown in Table S3, models employing either ECFP or the unique atom and coordination count feature set exhibit significantly higher accuracy compared to those using physicochemical features. Although ECFP offers a slight performance improvement over the unique atom and coordination count feature set, its large number of features and lack of interpretability make it less suitable for a small dataset (i.e., 28 experimental measurements). Therefore, we believe that our chosen feature set-counting unique atoms and their coordination-provides the best balance between model accuracy and interpretability.

We have added more details to the manuscript to further clarify our choice of features:

“This approach allows us to effectively capture the structural diversity of additives while maintaining interpretability, which is particularly important given the small size of the dataset. Additionally, as demonstrated in our benchmarking analysis (Table S3), this feature set offers a good balance between model accuracy and complexity, making it well-suited for identifying structure-property relationships in the given context.”

Table S3 Prediction accuracy of GPR models using different feature sets. Training and testing were repeated 100 times using different random states.

Test MAE*	Physicochemical feature	Extended connectivity fingerprint	Unique atom and coordination count (This work)
ASI	27.0 ± 9.4	19.4 ± 8.6	20.2 ± 6.9
ΔASI	20.9 ± 6.3	14.8 ± 7.5	16.7 ± 6.1
Q	16.7 ± 4.4	11.8 ± 4.8	12.2 ± 4.3
No. of features	66	2048	24

*Train:test = 80:20

We conducted feature importance analysis using Shapley Additive Explanation (SHAP). The Shapley values for the 10 most important features influencing the model predictions of ASI, ΔASI, and Q are presented in Fig. S18. The results show similar trends to those observed in the Spearman correlation analysis (Fig. 4b, c, d). For instance, in predicting ASI, the C_2_inRing feature is positively correlated with the model output, while the B[-1]_4_inRing and P[-1]_6_inRing features have a negative impact on the predictions.

We have added the use of SHAP analysis in the manuscript:

“To better understand the impact of each feature on model predictions, we performed feature importance analysis using Shapley Additive Explanation (SHAP). The SHAP analysis revealed the contribution of the most important features to the predictions of ASI, Δ ASI, and Q, providing insights that aligned well with the trends observed in the Spearman correlation analysis (Fig. S18). For instance, in predicting ASI, the C_2_inRing feature is positively correlated with the model output, while the B[-1]_4_inRing and P[-1]_6_inRing features have a negative impact on the predictions. This further supports the interpretability and reliability of our models.”

Figure S18 SHAP analysis of model features with respect to the model predictions of ASI, Δ ASI and Q.

3. The study appears to use exploitation to validate the ML model's predictive capability with 128 sets. However, Bayesian based active learning, which is more effective and has been confirmed its validity from various researches, should be considered for the optimization process.

We appreciate the reviewer’s suggestion. In addition to exploitation, we have now incorporated Bayesian optimization to predict potential additive systems for experimental validation. Using various acquisition functions, including upper confidence bound, probability of improvement, and expected improvement, we identified the top six predicted additive candidates, which are listed in Table S5. Notably, the top four candidates, based on fitness ranking, were identical for both exploitation and Bayesian optimization. The two additional additive systems suggested by Bayesian optimization—LiDFOB@1.0 wt% + TMSPA@1.0 wt% and LiBOB@1.0 wt% + TMSPA@1.0 wt%—have had their performance metrics (ASI, Δ ASI, and Q) experimentally measured, and the results are provided in Table S6. Since we were not able to use the same cathode composition for these additives (due to materials shortage), it is not possible to compare their measured performance data (Table S6) directly with previous results (Table 1). Nevertheless, we found LiDFOB@1.0wt%+TMSPa@1.0wt% promising as it outperforms the baseline in all three metrics.

Table S5 Top additive systems predicted using Exploitation and Bayesian optimization

Fitness rank	Exploitation	Bayesian optimization		
		Upper Confidence Bound	Probability of Improvement	Expected Improvement
1	LiDFOB_1+SA_1	LiDFOB_1+SA_1	LiDFOB_1+SA_1	LiDFOB_1+SA_1
2	LiBOB_1+TMSPi_1	LiBOB_1+TMSPi_1	LiBOB_1+TMSPi_1	LiBOB_1+TMSPi_1
3	LiBOB_1+MS_1	LiBOB_1+MS_1	LiBOB_1+MS_1	LiBOB_1+MS_1
4	LiBOB_1+SA_1	LiBOB_1+SA_1	LiBOB_1+SA_1	LiBOB_1+SA_1
5	VC_1+SA_1	VC_1+SA_1	VC_1+SA_1	LiDFOB_1+TMSPa_1
6	LiBOB_1+LDF_1	LiDFOB_1+TMSPa_1	LiDFOB_1+TMSPa_1	LiBOB_1+TMSPa_1

Table S6 Experimentally validation of the newly predicted formula of LiBOB_1+TMSPa_1, LiDFOB_1+TMSPa_1 compared with the baseline. Note this laminate composition (A-C024) is different than the previous laminate (LN210035-179-1) used for the previous additive testing, therefore this data cannot be used directly to compare with the previous data, but the three formulations are comparable with each other.

	ASI	Δ ASI	Q
Baseline	44.54	21.17	72.11
LiBOB_1+TMSPa_1	37.15	46.59	67.75
LiDFOB_1+TMSPa_1	34.79	21.02	77.89

Cathode: LN210035-179-1 (single-sided)	Cathode: A-C024 (single-sided; calendered)
90 wt% Targray LNMO "5V spinel"	90 wt% Targray LNMO "5V spinel"
5.69 wt% Timcal C-45	5.69 wt% Timcal C-45
0.05 wt% Tuball SWCNT	0.05 wt% Tuball SWCNT
4.26 wt% solvay 5130 PVDF Binder	4.26 wt% Solvay 5130 PVDF Binder
•SS- single sided ->calendered	Al Current Collector Thickness: 17 μ m
Total Electrode Thickness 83 μ m	Total Electrode Thickness: 79 μ m
Porosity: 40.9 %	SS Coating Thickness: 62 μ m
Total SS Coating Loading 15.21 mg/cm ²	Porosity: 37.6 %
Total SS Coating Density, 2.31 g/cm ³	Total SS Coating Loading: 15.09 mg/cm ²
Estimated C/10 areal capacity 1.85 mAh/cm ²	Estimated C/10 Areal Capacity: 1.81 mAh/cm ²

Fig. 3. Specific capacity of LNMO//Gr using newly predicted formula of LiBOB_1+TMSPa_1, LiDFOB_1+TMSPa_1 compared with the baseline

We have added the following sentence to the manuscript:

“Although our approach focused on exploitation, the top candidates identified were largely consistent with those obtained through Bayesian optimization methods (see SI and Table S5 for more details).”

4. The rationale for using ML in this study is unclear. While conducting 128 experiments is time-consuming and costly, the search space is relatively small from an ML perspective. The authors should justify that 128 sets are sufficient and that the best result is included. A more rigorous search and justification for the additive combinations are needed. It seems conducting 128 experiments would be more effective.

We agree with the reviewer that the search space of 125 additive combinations is relatively small from an ML perspective, considering the broad possibilities for varying the concentration of individual components. Instead, we focused on an initial set of 140 combinations derived from mixing 14 cathode additives with 10 anode additives, each at an equal weight percentage of 1%. We generated the unknown compositions in this way for two reasons: (1) their alignment with the training dataset, in which 15 out of 28 additives consisted of both a cathode and an anode additive at 1%, minimizing extrapolation in the composition space; and (2) empirical evidence supporting 1% as an effective concentration for various anode and cathode additives.(Tornheim et al., 2017)

While conducting 125 experiments would yield most accurate results, it is important to note that each additive requires triplicate cells, and each cell must be tested using a dedicated MACCOR channel. Besides being time-consuming, the need for 375 (125 × 3) MACCOR channels with temperature control would drive the equipment cost to over \$500,000, without reducing the testing duration.

We have modified the manuscript to further clarify the rationale behind our search space generation and ML usage:

“...We generated the unknown compositions in this way for two reasons: (1) alignment with the training dataset, in which 15 out of 28 combinations consisted of both additives at 1%, minimizing extrapolation in the composition space; and (2) empirical evidence indicating that 1% is an effective concentration for various anode and cathode additives.(Tornheim et al., 2017) We noted that brute-force experimental screening of all 125 additive combinations would be highly inefficient in terms of both time and cost (see SI for more details)”

5. The extrapolation performance of the ML model should be validated, as ML models are typically weak in this area. Dimensionality reduction techniques should be applied to both train (28) and test (128) sets to understand the ML search space and determine whether it involves extrapolation or interpolation.

Following the reviewer’s suggestion, we conducted a principal component analysis (PCA) on all considered additives and projected their features onto the first two principal components, as shown in Fig. S19. The results indicate that the features of the initial 28 additives are diverse enough to overlap with approximately 85 out of the 125 unknown additives, suggesting reasonable coverage of the feature space. This overlap provides confidence that the model, trained on the initial 28 additives, has sufficient representation of the feature space to make reliable predictions for the majority of the 125 unknown additive systems.

We have modified the manuscript accordingly to reflect this observation:

“Principal component analysis suggests that the model trained on the initial 28 additives adequately represents the feature space, allowing for reliable predictions for most of the 125 unknown additive systems (Fig. S19)”

Figure S19 Graphical illustration of the feature space of all considered additives projected onto principal component 1 and 2.

6. The authors should demonstrate that the optimal set of additives is also effective for different chemical compositions of LNMO (e.g., $\text{LiNi}_{0.6}\text{Mn}_{1.4}\text{O}_4$ or $\text{LiNi}_{0.5}\text{Mn}_{1.4}\text{Dopant}_{0.1}\text{O}_4$), as the tested LNMO composition may not be the best cathode material.

We appreciate the reviewer's suggestions. The uniqueness of additives lies in their primary role at interfaces; through enhancing interfacial stability, particularly at the cathode electrolyte interface (CEI) and solid electrolyte interface (SEI), they help achieve lower resistance (ASI), reduced lithium inventory loss, and minimized transition metal dissolution. As shown in the supplementary information, the main performance improvements are attributed to mitigating lithium inventory loss, which is the primary degradation mechanism, alongside associated transition metal dissolution and high impedance rise. Additives are typically designed for specific tasks, such as SEI formation on the anode, CEI formation on the cathode, and unique applications like fast-charging and extreme-temperature performance. Researchers from the Dahn group and Argonne National Laboratory have been extensively developing both singular and combinatorial additives. Given that the surface chemistry of $\text{LiNi}_{0.6}\text{Mn}_{1.4}\text{O}_4$ or $\text{LiNi}_{0.5}\text{Mn}_{1.4}\text{Dopant}_{0.1}\text{O}_4$ is like that of LNMO, the application of ML for optimizing additives for LNMO would be equally applicable to $\text{LiNi}_{0.6}\text{Mn}_{1.4}\text{O}_4$ and $\text{LiNi}_{0.5}\text{Mn}_{1.4}\text{Dopant}_{0.1}\text{O}_4$ systems. In our previous research, changes in TM composition were shown to alter the optimal species and concentrations of additives. (Peebles et al., 2017; Yang et al., 2021; Yang et al., 2022) However, determining the precise additive formula

would require an experimental dataset for these cathodes using the same ML methodology, which lies beyond the scope of this paper.

Reviewer #2 (Remarks on code availability):

It seems the results are reproducible based on the provided codes.

References

- Batteries*. (2021). In C. Liao (Ed.), *Materials principles and characterization methods*. doi:10.1088/978-0-7503-2682-7
- Peebles, C., Sahore, R., Gilbert, J. A., Garcia, J. C., Tornheim, A., Bareño, J., . . . Abraham, D. P. (2017). Tris(trimethylsilyl) Phosphite (TMSPi) and Triethyl Phosphite (TEPi) as Electrolyte Additives for Lithium Ion Batteries: Mechanistic Insights into Differences during LiNi_{0.5}Mn_{0.3}Co_{0.2}O₂-Graphite Full Cell Cycling. *Journal of The Electrochemical Society*, *164*(7), A1579. doi:10.1149/2.1101707jes
- Tornheim, A., Peebles, C., Gilbert, J. A., Sahore, R., Garcia, J. C., Bareño, J., . . . Abraham, D. P. (2017). Evaluating electrolyte additives for lithium-ion cells: A new Figure of Merit approach. *Journal of Power Sources*, *365*, 201-209. doi:<https://doi.org/10.1016/j.jpowsour.2017.08.093>
- Yang, J., Fonseca Rodrigues, M.-T., Son, S.-B., Garcia, J. C., Liu, K., Gim, J., . . . Liao, C. (2021). Dual-Salt Electrolytes to Effectively Reduce Impedance Rise of High-Nickel Lithium-Ion Batteries. *ACS Applied Materials & Interfaces*, *13*(34), 40502-40512. doi:10.1021/acsmi.1c08478
- Yang, J., Rodrigues, M.-T. F., Yu, Z., Son, S.-B., Liu, K., Dietz Rago, N. L., . . . Liao, C. (2022). Design of a Scavenging Pyrrole Additive for High Voltage Lithium-Ion Batteries. *Journal of The Electrochemical Society*, *169*(4), 040507. doi:10.1149/1945-7111/ac613f

Reviewer #1 (Remarks to the Author):

The resubmitted manuscript shows that the authors have made revisions and offered supplementary materials as much as possible in response to the comments. Overall, the response and revision are satisfactory.

We thank the reviewer for positive comments.

Nevertheless, I still have a short comment on the response to Reviewer1_Question1 for your reference as below, even though the question isn't decisive to the article acceptance. There is a significant correlation between the selected performance indicators in the study with Pearson's correlation coefficients of 0.93, -0.79 and -0.87 for ASI/ Δ ASI, ASI/Q and Δ ASI/Q, respectively. The authors believe that the correlation indicator suggests that it is possible to consider the use of only one indicator. Although the absolute value of the coefficients is close to 1, a larger number of factors/indicators still need to be considered. We can assume that all performance indicators form a vector space and that the Δ ASI, ASI, and Q indicators are linear combinations of similar basis vectors in that space, and therefore have some correlation. However, these combinations also contain different components that play an important role in the discovery of new electrolytes, making it necessary to give due consideration to additional performance indicators.

We appreciate the reviewer's suggestion and agree that incorporating additional performance indicators beyond ASI, Δ ASI, and Q could provide valuable insights for identifying new and ideal electrolytes. However, it is also important to weigh the trade-offs, as each additional objective increases the complexity of the optimization algorithm and the associated data requirements. We believe that the three objectives utilized in this study achieve an effective balance between optimization efficiency and practical applicability for additive discovery.

We have revised our manuscript as follows:

“Although other parameters, such as initial capacity (Q_{ini}) and coulombic efficiency (CE) during cycling, can also indicate battery performance, ASI, Δ ASI, and Q are the dominant factors showing the effect of additives on high-performance lithium-ion batteries. **Furthermore, we believe that leveraging these three objectives provides an effective compromise between optimization efficiency and practical utility, ensuring a robust approach to additive discovery.**”

Reviewer #1 (Remarks on code availability):

The code can be run well.

We thank the reviewer for reviewing our code.